# Role of Cannabinoid CB2 Receptor in Alcohol Use Disorders: From Animal to Human Studies

**DOI:** 10.3390/ijms23115908

**Published:** 2022-05-25

**Authors:** María Salud García-Gutiérrez, Francisco Navarrete, Ani Gasparyan, Daniela Navarro, Álvaro Morcuende, Teresa Femenía, Jorge Manzanares

**Affiliations:** 1Instituto de Neurociencias, Universidad Miguel Hernández-CSIC, Avda de Ramón y Cajal s/n, San Juan de Alicante, 03550 Alicante, Spain; maria.ggutierrez@umh.es (M.S.G.-G.); fnavarrete@umh.es (F.N.); agasparyan@umh.es (A.G.); dnavarro@umh.es (D.N.); amorcuende@umh.es (Á.M.); tfemenia@umh.es (T.F.); 2Red Primaria de Adicciones, Instituto de Salud Carlos III, MICINN and FEDER, 28029 Madrid, Spain; 3Instituto de Investigación Sanitaria y Biomédica de Alicante (ISABIAL), Hospital General Universitario de Alicante, Centro de Diagnóstico 5ª Planta (Edf. Gris), Avda. Pintor Baeza 12, 03010 Alicante, Spain

**Keywords:** cannabinoid CB2 receptors, CB_2_KO, alcohol, reward system, pharmacological studies

## Abstract

Cumulative evidence has pointed out cannabinoid CB2 receptors (CB_2_r) as a potential therapeutic key target for treating alcohol use disorder (AUD). This review provides the most relevant results obtained from rodent and human studies, including an integrative section focused on the involvement of CB_2_r in the neurobiology of alcohol addiction. A literature search was conducted using the electronic databases Medline and Scopus for articles. The search strategy was as follows: “Receptor, Cannabinoid, CB2” AND “Alcohol-Related Disorders” AND “human/or patients”; “Receptor, Cannabinoid, CB2” AND “Alcohol” OR “Ethanol” AND “rodents/or mice/or rats”. Pharmacological approaches demonstrated that the activation or blockade of CB_2_r modulated different alcohol-addictive behaviors. Rodent models of alcoholism revealed significant alterations of CB_2_r in brain areas of the reward system. In addition, mice lacking CB_2_r (CB_2_KO) show increased alcohol consumption, motivation, and relapse alterations. It has been stressed that the potential neurobiological mechanisms underlying their behavioral effects involve critical elements of the alcohol reward system. Interestingly, recent postmortem studies showed *CNR2* alterations in brain areas of alcoholic patients. Moreover, although the number of studies is limited, the results revealed an association between some genetic alterations of the *CNR2* and an increased risk for developing AUD. This review provides evidence that CB_2_r may play a role in alcohol addiction. Clinical studies are necessary to figure out whether CB_2_r ligands may prove useful for the treatment of AUD in humans.

## 1. Introduction

Alcohol use disorder (AUD) represents the seventh most common leading risk factor for premature death and disability worldwide, accounting for more than 3 million deaths annually (5.3% of all deaths) [1]. The situation worsens in the age range of 15–49 years, being the leading cause of disability. Comorbidity with psychiatric disorders occurs as alcohol dependence progresses, significantly increasing disability, morbidity, and mortality rates associated with AUD [2]. Despite these devastating data, treatment options for treating AUD are scarce and present limited efficacy. Currently, there are only four medications approved with different mechanisms of action: acamprosate, disulfiram, naltrexone, and nalmefene [3,4]. These drugs, especially naltrexone and acamprosate, are effective in certain patients. However, it is disturbing that approximately 70% of patients relapse within the first year of treatment [5,6]. Unfortunately, the FDA and EMA has approved no new drugs for almost the last decade.

The development of new and effective treatment options for AUD encompasses identifying the underlying neurobiological mechanisms that are still pending despite the efforts made. In this respect, a growing and compelling body of evidence shows that cannabinoid receptors (CB_1_r and CB_2_r) play a relevant role in AUD. Cumulative pieces of evidence have supported the role of CB_1_r in the regulation of ethanol intake, the establishment of tolerance/dependence, and the vulnerability to relapse, mainly through animal studies [7,8,9,10,11]. Consequently, pharmacological modulation of CB_1_r employing CB_1_r-agonists or -antagonists attracted strong interest due to its therapeutic potential [9,12,13,14,15,16]. However, CB_1_r agonists have been associated with essential side effects, such as psychotomimetic effects and abuse and dependence problems [17,18]. In addition, increased anxiety and suicidal ideation rates were observed in obese patients treated with the CB_1_r-antagonist SR141716A, marketed as rimonabant, reasons for which it had to be withdrawn from the market [19,20,21,22]. These facts led to the exploration of additional lines of research, among which CB_2_r highlights for lacking such undesirable effects and being involved in the addictive properties of alcohol.

CB_2_r, initially considered the peripheral cannabinoid receptor, has been expressed in the central nervous system, playing a relevant role in modulating several processes [23,24,25,26,27,28,29,30]. This cannabinoid receptor is a G protein-coupled receptor that inhibits adenylyl cyclase’s activity through its Gi/Goα subunits [31,32]. Interestingly, in some cells, such as human leukocytes, CB_2_r are coupled to Gα_s_ activating, in this case, the adenylyl cyclase and, therefore, increasing cAMP [33,34]. Indeed, CB_2_r are also known to activate the MAPK-ERK pathway through Gβγ subunits, inducing changes in cell migration [35,36,37].

This cannabinoid receptor CB_2_r is widely expressed through the brain, from cortical areas to the cerebellum, including limbic regions, such as the hippocampus (HIP) and amygdala (AMY), and regions within the reward system, such as the ventral tegmental area (VTA) and nucleus accumbens (NAcc) [23,25,38,39]. Despite some controversy, the progress in antibodies against CB_2_r have allowed identifying the expression of this cannabinoid receptor in neurons within the prefrontal cortex, dorsal striatum, NAcc, VTA, HIP, and cerebellum [28,29,38,39,40,41,42,43,44]. In more detail, CB_2_r is expressed in dopaminergic [38,42,45,46,47,48], glutamatergic [46,49], and GABAergic neurons [50,51]. In addition, CB_2_r has been identified in astrocytes [38,52,53] and microglia [44,54,55,56], in both basal and activated states, downregulating the gene expression of inflammatory mediators [57,58,59]. The expression of CB_2_r in brain regions belonging to classical neuronal circuits involved in drug addiction, such as the VTA, NAcc, AMY, and HIP, has promoted further studies to understand its potential role in these pathological conditions.

This review is aimed at providing information from animal studies, including pharmacological and genetic manipulations, and human studies, supporting the role of CB_2_r in alcohol addiction. Furthermore, we are reviewing the neurobiological mechanisms underlying the modulatory effects of CB_2_r on alcohol addiction. Finally, we provide concluding remarks highlighting the next steps needed for achieving the goal that compounds acting on CB_2_r from entering into clinical practice soon.

## 2. Results

### 2.1. Evidence from Rodent Studies

#### 2.1.1. Changes of CB_2_r Induced by Ethanol in Brain Regions of the Reward System

The involvement of CB_2_r in AUD has been studied in several animal models simulating different stages of alcohol addiction. The reduction in the gene encoding for CB_2_r (*CNR2*) gene expression in the ventral midbrain of mice developing increased ethanol preference was the first evidence of functional changes in this receptor. In contrast, no alteration of *CNR2* was found in the ventral midbrain of mice with lower preference (Table 1) [60].

Since then, changes in CB_2_r have been detected in brain regions of rodents exposed to different patterns of ethanol consumption. Sanchez-Marin et al. [63] identified a significant reduction of *CNR2* expression in the striatum and HIP of adult rats exposed to a forced intermittent ethanol intoxication (binge drinking) (3 g/kg; i.p.; 4 consecutive days × 4 weeks) during adolescence. These results highlight the involvement of CB_2_r in the long-lasting emotional disturbances (increased anxiety-like behaviors and cognitive impairments) induced by ethanol intoxication during adolescence. Later, the same authors identified a significant increase and decrease of *CNR2* expression in the HIP and AMY, respectively, of adult rats exposed to an intermittent voluntary ethanol consumption during adolescence (EtOH 10% (*v/v*); 4 days/24 h/4 w). These alterations were accompanied by an increase of glial fibrillary acidic protein (GFAP) and allograft inflammatory factor 1 (AIF-1) in the medial prefrontal cortex (mPFC) and HIP, respectively, crucial brain areas for the development of alcohol addiction [64].

A recent study developed by Ledesma et al. [61] has focused on analyzing the impact of ethanol binge drinking during adolescence on cognitive disturbances. This study showed that pre-exposure to ethanol (2.5 g/kg, i.p.) during adolescence increased *CNR2* and CB_2_r protein expression in the HIP of wild-type (WT) mice at 6- and 12-months of age. Interestingly, in the transgenic mice model of Alzheimer’s disease, APPswe/PS1dE9 mice, *CNR2* expression reduced at 6-months of age while statistical significance was notified at 12-months. The authors concluded that the increased levels of CB_2_r in WT could be a protective mechanism for ethanol-induced cognitive impairments since WT-EtOH mice at 6-months of age did not display learning impairments. On the contrary, the reduction of CB_2_r in APPswe/PS1dE9 mice appears to be a negative hallmark for displaying cognitive impairments.

More recently, CB_2_r receptor availability and its correlation with ethanol-induced locomotor sensitization have been evaluated. For this purpose, mice were treated with ethanol (EtOH, 2 g/kg/d; i.p.) for 21 days and classified into EtOH-high or -low groups, according to their behavioral variability on locomotor activity on the 21st day of acquisition. A significant decrease in CB_2_r protein expression was detected in the infralimbic cortex and HIP (CA1, CA2, and CA3 fields) of the EtOH-low group. Interestingly, mice were subjected to 5 days of EtOH withdrawal in an additional independent experiment after completion of the acquisition phase. An increase of CB_2_r protein levels was found in the basolateral nucleus of the amygdala (BLA) and the nucleus accumbens shell (Acbsh) (both groups), in the anterior cingulate cortex (CG1) and nucleus accumbens core (Acbco) (only in EtOH-low), and CeA (central nucleus of amygdala) and dorsolateral striatum (Dis) (only in EtOH-high). The authors concluded that CB_2_r upregulation might be correlated with withdrawal aspects of EtOH-sensitized mice, pointing out the need for further studies to clarify this aspect. In the third set of experiments, changes in CB_2_r protein levels were analyzed in mice injected with EtOH (1.4 g/kg) (EtOH challenge) after the withdrawal period. In this case, CB_2_r were downregulated in the AMY and striatum of both EtOH groups, regardless of motor sensitization. Moreover, in all experimental groups (including controls), CB_2_r was reduced in the prefrontal cortex and HIP. The results showed that CB_2_r downregulation appeared to be related to the development of motor sensitization [62].

Additional studies evaluated *CNR2* alterations induced by ethanol withdrawal. Serrano et al., [65] studied the effects of single or repeated-ethanol withdrawal periods on *CNR2* in the AMY of rats. Briefly, rodents were exposed to a chronic continuous ethanol diet (10% *w/v*; 15 days) or an intermittent ethanol consumption (10% *w/v*; 5-days/week × 3 weeks), evaluating changes in *CNR2* expression at 6 h and 24 h after last ethanol exposure. No significant alterations were found in either group’s *CNR2* expression at 6 h. Surprisingly, a small but non-significant reduction was observed at 24 h in rats exposed to the continuous ethanol diet, and a statistically significant decrease in the ethanol intermittent consumption group. These results revealed that exposure to repeated withdrawal periods induced more broad and robust alterations in *CNR2* in the AMY, suggesting its potential implication in negative motivational states and enhanced stress responsivity associated with alcohol dependence and withdrawal.

Epidemiological data and clinical and preclinical studies indicate that exposure to stressful situations increases vulnerability to alcohol consumption. Although several studies have assessed this correlation, only a few have examined the involvement of CB_2_r. In this respect, Marco et al. [66] identified reduced levels of CB_2_r protein expression in the hippocampal formation of female Wistar rats with high ethanol consumption and preference after previous exposure to chronic mild stress (CMS) during 6 weeks. No changes were observed in the frontal cortex. Interestingly, male Wistar rats exposed to the same paradigm did not display ethanol preference nor increased ethanol consumption, suggesting significant sex-dependent vulnerability to ethanol after CMS model exposure. Additionally, other studies demonstrated that the combination of restraint stress with intermittent alcohol exposure (3 g/kg ethanol; p.o.; 4 days/week for 4 weeks) during adolescence significantly increases CB_2_r in the AMY of young adults [67].

In summary, evidence achieved to date showed that ethanol modified the gene and protein expression of CB_2_r across the brain. Some of these studies obtained opposite changes suggesting that several factors, such as the strain of rodent, the pattern (acute vs. chronic) and route (oral, i.p., etc.) of ethanol administration, the animal model (conditioned place preference (CPP), ethanol-self administration), and the techniques used to measure *CNR2* and CB_2_r protein expression, may influence the alterations observed. Moreover, advanced techniques, such as single-cell gene expression, RNAscope in situ hybridization (ISH), and immunohistochemistry (IHC) would be of great interest to further characterize cell-specific changes of *CNR2* and CB_2_r protein expression induced by ethanol in neurons, microglia, and astroglia.

#### 2.1.2. Genetic Studies in Rodents

Further support of the role of CB_2_r on the vulnerability to ethanol consumption has been obtained using genetically-modifies mice.

On the one hand, mice lacking CB_2_r (CB_2_KO) showed higher sensitivity to withdrawal, as indicated by an increased handling-induced convulsion score after an acute high dose of ethanol (Table 2). Moreover, CB_2_KO mice presented higher voluntary ethanol intake and preference, ethanol-conditioned place preference, increased motivation to drink, and higher levels of ethanol consumption in the oral ethanol self-administration paradigm. These behavioral alterations were accompanied by significant changes in the gene expression of critical targets underlying alcohol addiction, such as the mu-opioid receptor (*Oprm1*), reduced in the NAcc of naïve CB_2_KO mice. Interestingly, acute ethanol administration increased tyrosine hydroxylase (*TH*) in the VTA at all doses tested in CB_2_KO mice. In contrast, a significant reduction was observed in WT at 0.5 and 1 g/kg of ethanol doses. Interestingly enough, *Oprm1* in the NAcc of CB_2_KO significantly increased after ethanol administration (1 and 2 g/kg). In contrast, no modification was observed in WT mice. The conclusion that can be drawn from this study is that deletion of CB_2_r influences ethanol-induced effects in *TH* and *Oprm1* in brain targets of the mesolimbic system that may contribute, at least in part, to the higher sensitivity to ethanol’s rewarding effects observed in CB_2_KO mice [68].

Similarly, in an additional study, the absence of CB_2_r in mice enhanced ethanol conditioning in the conditioned place preference (CPP) paradigm. However, no changes were observed in the two-bottle choice paradigm between CB_2_KO and WT mice [69]. These results contrast with those provided by Ortega-Alvaro et al. [68]. Such discrepancies could be related to the genetic background strain of the KO mice used (C57Bl/6J and CD1, respectively). Furthermore, Powers and colleagues carried out the two-bottle choice paradigm 3 weeks after the CPP, that modified CB_2_r expression in WT due to the repeated ethanol exposure in CPP. These modifications may explain why no differences between the CB_2_KO and WT mice were observed in the CPP by Powers et al. [69].

Likewise, the involvement of CB_2_r in modulating the effects of stressful situations on alcohol consumption has been addressed by additional studies. In this respect, Pradier et al. [70] aimed to determine if CB_2_r has a role in ethanol consumption induced by social isolation. To this aim, ethanol intake was evaluated in CB_2_KO and WT mice housed individually or in the group. Both strains did not differ in the forced drinking paradigm, i.e., ethanol intake was lower in individually housed mice when compared to grouped mice, regardless of the genotype. However, apparent differences were observed in the intermittent forced drinking model. CB_2_KO grouped and individually housed mice did not differ in ethanol intake whereas for WT ethanol consumption was higher in individually housed mice when compared to grouped animals. Altogether, these results supported the modulatory function of CB_2_r in the social environmental-induced alcohol intake.

Moreover, the involvement of CB_2_r in modulating the impact of acute stress exposure on ethanol consumption has also been addressed using a knockout mouse lacking the CB_2_r in midbrain dopaminergic neurons (DAT-*CNR2* conditional knockout mice). Under basal conditions, alcohol preference was lower in DAT-CNR2-/- mice compared with controls. Interestingly, these differences were more pronounced when mice were exposed to an acute stress protocol. Control mice showed a significant enhanced preference in the CPP paradigm that was not present in DAT-*CNR2*-/- mice. These results supported the idea that the absence of CB_2_r in dopaminergic neurons decreases alcohol consumption, even after acute stress, supporting the involvement of CB_2_r in alcohol dependence [71].

Together, these studies show that CB_2_r regulates rewarding ethanol effects since full CB_2_KO mice showed increased ethanol preference, motivation and consumption in different animal models. Moreover, full and conditioned CB_2_KO mice revealed that CB_2_r also modulates the impact of stress exposure on ethanol intake.

#### 2.1.3. Pharmacological Studies in Rodents

To date, several pharmacological studies have been carried out in rodents to assess how the activation or blockade of CB_2_r modulates alcohol consumption, motivation, and relapse (Table 3). The first study revealed that systemic CB_2_r-agonist JWH015 (20 mg/kg, i.p.) administration induced a marked increase in ethanol intake in mice previously exposed to the CMS model. Interestingly, the CB_2_r-antagonist AM630 (3 mg/kg, i.p.) induced just the opposite effects, preventing the development of alcohol preference (Table 3) [60].

Similarly, Al Mansouri et al. [72] demonstrated that β-caryophyllene, a selective CB_2_r-agonist, significantly reduced voluntary alcohol consumption and CPP in male C57BL/6J mice. These effects were blocked by the administration of the antagonist AM630, supporting the role of CB_2_r as the target involved in the effects of β-caryophyllene.

Likewise, Liu et al. [73] showed that JWH133 (5 mg/kg, i.p.) completely blocked alcohol-induced CPP. Similar effects occurred following the administration of the CB_2_r-agonist, JWH133, which reduced alcohol and food rewarding behaviors in C57BL/6 mice during the acquisition phase [75]. In contrast, Powers et al. [69] failed to observe any effect on alcohol consumption and CPP in HS/Ibg mice treated with CB_2_r-agonist JWH133 (10 and 20 mg/kg; i.p.) or CB_2_r-antagonist AM630 (10 and 20 mg/kg; i.p.). Discrepancies regarding the effects of pharmacological regulation of rCB_2_r on alcohol consumption may be due to various experimental circumstances, such as using different mouse strains (HS/Ibg or C57BL/6J) or the dose, route, or schedule of administration of cannabinoid compounds.

Pharmacological modulation of CB_2_r by the antagonist AM630 (1 mg/kg, i.p.) significantly increased the motivation to drink and the level of alcohol consumption in the oral operant self-administration paradigm [74]. Furthermore, this study showed how the agonist JWH133 (1 mg/kg, i.p.) produced opposite effects, significantly reducing alcohol self-administration. These changes were accompanied by bidirectional modifications in the gene expression of brain targets related to regulating the reinforcing effects of alcohol, such as *TH* in the VTA and *Oprm1*, *CNR1* and *CNR2* in the NAcc (for details, see below).

In another experimental paradigm in which rats were fed with liquid diets of ethanol (10%) for 2 weeks, treatment with JWH133 specifically counteracted the impairment of adult neural progenitor cell (NPC) proliferation induced by ethanol. In detail, JWH133 increased the number of BrdU+ cells in the subgranular zone of the dentate gyrus, subventricular area of the lateral ventricles, and hypothalamus [76].

Indeed, a recent study revealed that the CB_2_r-agonist JWH133 (0.2 mg/kg) increased astrocytes and microglia in the HIP of rats exposed to a model of subchronic (2 weeks) ethanol diet (11% *v/v*) exposure. These results suggested a role of CB_2_r in astrocytes and microglial recruitment and activation following specific neurotoxic stimuli [77].

Altogether, the majority of the pharmacological studies carried out to date revealed that the agonism of CB_2_r can reduce ethanol consumption, preference, and motivation to drink in several animal models. Moreover, CB_2_r activation is proposed to induce a neuroprotective effect against ethanol neurotoxicity, through its modulatory effects on microglia and astroglia. Interestingly, key targets of the opioidergic and dopaminergic systems would play a role in such effects.

### 2.2. Evidence from Human Studies

The role of CB_2_r in AUD in humans has not been extensively studied. Onaivi’s group provided the first reference [60]. This study described for the first time an association between the Q63R polymorphism in the *CNR2* (glutamine in amino acid position 63 substituted by arginine) and alcoholism in a population of Japanese patients. In short, this study indicated that this single nucleotide polymorphism (SNP) might be a risk factor for alcoholism, at least in Japanese people. More recently, one genome-wide association study (GWAS) found a positive association cluster around the 3′ untranslated region (3′UTR) of the *CNR2* and AUD [73].

Interestingly, our group recently carried out a postmortem study analyzing *CNR2* in the dorsolateral prefrontal cortex (DLPFC) and NAcc of patients with alcohol dependence (unpublished data), showing an observed reduction in both regions. Conversely, another study found an increase *CNR2* in human monocyte-derived dendritic cells from alcohol users [78].

Altogether, these data suggested that CB_2_r is involved in AUD in humans and emphasized the need for future large-scale studies of alcohol-dependent patients to determine its potential therapeutic role in AUD.

### 2.3. Involvement of CB_2_r in the Neurobiology of Alcohol Addiction

#### 2.3.1. CB_2_r and the Alcohol Brain’s Reward System

Drugs of abuse, such as alcohol, activate the mesolimbic dopaminergic system that mediates reinforcing and motivational actions [79]. Mainly, the rewarding effects of alcohol occurred by the interaction between the opioidergic and dopaminergic mesolimbic systems, involving the increased firing of dopamine neurons in the VTA [74] and a subsequent increase of dopamine release into the NAcc [80]. Furthermore, other systems, such as the cholinergic glutamatergic and GABAergic systems and molecular targets involved in neuroinflammation, have been proposed to display a relevant role in alcohol actions. The following sections describe the available information on the role of CB_2_r in regulating dopaminergic, opioidergic and cholinergic neurotransmissions in the mesocorticolimbic circuit, neuroinflammatory mediators activated by alcohol consumption and excitatory-inhibitory balance, and, consequently, its involvement in the modulation of alcohol rewarding properties.

#### 2.3.2. CB_2_r and the Dopaminergic System

A crucial step to deepen the functional role of CB_2_r in alcohol reward was to improve the understanding of its interaction with the dopaminergic system. Double confocal immunohistochemical analyses revealed that CB_2_r are expressed in neurons and astrocytes of the NAcc and VTA of CD1 male mice [38,81]. Similarly, Zhang et al. confirmed these findings by in situ hybridization and immunohistochemical assays detecting *CNR2* gene expression and CB_2_r immunostaining in VTA dopaminergic neurons [46]. CB_2_r was also found in VTA dopaminergic neurons of WT mice, whereas it was absent in conditional knockout mice in which *CNR2* was selectively suppressed in these neurons (DAT-Cnr2 cKO mice) [73].

CB_2_r immunoreactive cells and dopamine D2 receptor (D2Dr) were present in the NAcc and VTA of WT mice [38]. Recently, CB_2_r was also identified in striatal medium spiny neurons that express dopamine D1 or D2 receptors by RNAscope ISH assay [82]. Interestingly, repeated CB_2_r activation induced a significant D2Dr upregulation by involving GRK5, β-Arrestin 2, and ERK1/2 protein signaling [83]. Altogether, these results suggest possible functional cooperation between CB_2_r and dopamine receptors (Figure 1).

In vivo microdialysis studies were performed to evaluate if the genetic manipulation of CB_2_r in mice could modulate extracellular dopamine levels in the NAcc of mice. Xi et al. revealed that the deletion of CB_2_r (CB_2_KO mice) was not associated with any change in basal concentrations of extracellular dopamine between WT and CB_2_KO mice [84]. On the other hand, delta-9-tetrahydrocannabinol (THC) produced a dose-dependent decrease in extracellular dopamine in the NAcc of WT mice. In contrast, the opposite effect was obtained in CB_2_KO mice, suggesting that activation of the CB_2_r inhibits DA release in the NAcc [80]. Our group also carried out microdialysis experiments to investigate the effect of CB_2_r overexpression (CB2xP mice) on NAcc dopamine release. No differences were present between WT and CB2xP mice [38].

Despite CB_2_r deletion or overexpression not modifying NAcc dopamine levels, systemic administration of the selective CB_2_r agonist, JWH133 (3, 10, 20 mg/kg, i.p.), significantly and dose-dependently lowered extracellular NAcc dopamine in WT mice, but not in CB_2_KO mice. In addition, this reduction was blocked by the selective CB_2_r antagonist AM630 (10 mg/kg, i.p.), suggesting that CB_2_r is involved in the inhibitory effect of JWH133 on DA release.

Interestingly, to determine if this inhibitory effect was mediated by activation of brain or peripheral CB_2_r, intranasal or intra-NAcc local administration of JWH133 was assessed. Both routes of administration produced a significant reduction in extracellular dopamine levels suggesting the participation of brain CB_2_r located in the NAcc [84]. On the other hand, systemic or local administration of a CB_2_r inverse agonist, Xie2-64, into the NAcc reduced extracellular dopamine levels in a dose-dependent manner in rats [85].

To further explore the CB_2_r-mediated modulation of the dopaminergic tone in the VTA, the effects of JWH133 administration on neuronal dopamine firing were evaluated by electrophysiological methods. For that purpose, three different approaches were used: (1) perforated patch-clamp recording in single dissociated VTA dopaminergic neurons; (2) cell-attached patch-clamp in slice preparations (ex vivo); and (3) single-unit recording in anesthetized mice (in vivo). Interestingly, under these three experimental conditions, JWH133 significantly reduced the neuronal firing, effects that were blocked by AM630 [39]. A recent study further explored the underlying mechanisms of CB_2_r-mediated VTA dopaminergic neuronal function regulation. Employing patch-clamp recording in mouse VTA slices and dissociated VTA dopaminergic neurons, Ma and colleagues suggested that CB_2_r modulates the excitability mainly through an intrinsic mechanism, including reducing intracellular cAMP while enhancing M-type K^+^ currents. They also pointed out that although JWH133 reduces presynaptic glutamate release probability and results in a decreased glutamatergic synaptic transmission in VTA dopaminergic neurons, this may play a marginal role in the reduction of neuronal firing since pharmacological blockade of synaptic transmission failed to prevent the JWH133-induced inhibitory effect [45].

Another experimental approach to evaluate how CB_2_r manipulation could modulate dopaminergic transmission is the analysis of TH gene expression changes, the rate-limiting enzyme of dopamine synthesis. CB2xP mice showed increased TH and higher dopamine transporter (DAT) gene expression levels in the VTA [38]. In contrast, CB_2_KO mice presented a significant decrease in TH gene expression [81]. Similarly, in DAT-*CNR2* cKO mice, TH protein levels were modified in midbrain areas [86]. Interestingly, acute ethanol administration downregulated TH gene expression (0.5, 1, and 2 g/kg, p.o.) in the VTA of WT mice but induced a remarkable upregulation in CB_2_KO mice (1 and 2 g/kg, p.o.) [68]. Furthermore, pharmacological blockade of CB_2_r with AM630 (1 mg/kg, i.p.) increased TH gene expression in the VTA, whereas the administration of JWH133 (1 mg/kg, i.p.) produced a downregulation [74].

Overall, there is sufficient evidence suggesting that functional regulation of the CB_2_r can modify dopaminergic activity in the mesolimbic system, partly responsible for modulating alcohol’s reinforcing and motivational effects.

#### 2.3.3. CB_2_r and the Opioidergic System

Crosstalk between the endocannabinoid and opioidergic system has been widely studied [87,88,89,90]. However, very sparce information is available regarding the specific interaction between opioid receptors and CB_2_r and its regulation of alcohol reward. In this respect, valuable information through genetic and pharmacological approaches has been provided. A significant decrease in the expression of the gene encoding for mu-opioid receptors (Oprm1) was seen in the NAcc of CB2xP mice [38], while the opposite effect was observed in the NAcc of CB_2_KO mice [68]. In addition, acute ethanol administration did not change Oprm1 gene expression in the NAcc of WT mice, whereas a very significant increase appeared in CB_2_KO mice (1 and 2 g/kg, p.o.). Interestingly, the activation of CB_2_r by JWH133 (1 mg/kg, i.p.) significantly decreased Oprm1 levels while the blockade with AM630 (1 mg/kg, i.p.) induced an upregulation [75]. Thus, it could be argued that a lower CB_2_r functionality (genetic deletion (CB2^−/−^) or pharmacological blockade (AM630)) leads to an increased expression of Oprm1. This effect might, at least in part, represent the basis of the higher vulnerability to alcohol reward. Future studies are needed to elucidate how CB_2_r interacts with the opioidergic system in the mesolimbic circuit and its involvement in regulating the reinforcing effects of alcohol.

#### 2.3.4. CB_2_r and the Cholinergic System

Alcohol, like nicotine, requires neuronal nicotinic acetylcholine receptors (nAChRs) to display its rewarding effects within the mesolimbic system. Yorgason et al. demonstrated that the release of dopamine in the NAcc by EtOH is mediated by cholinergic interneurons and atypical GABA_A_ receptors [91]. Interestingly, the same group reported that low ethanol concentrations induced positive allosteric effects at nAChRs, which was correlated with an increase in spontaneous DA release [92]. The fact that both drugs share the latter molecular pathway has been proposed to contribute to the increased comorbidity of alcohol and nicotine dependence [93].

A close interaction between CB_2_r and nAChRs has been identified in different animal models of nicotine exposure and reward. One of the first studies evaluating the role of cannabinoid receptors in nicotine-induced discriminative behavior demonstrated that neither CB1r nor CB_2_r were involved [94]. Some years later, the actions of cannabinoid receptor activation by WIN55,212-2, a CB_1_/CB_2_r agonist, on nicotine self-administration were explored. WIN55,212-2 significantly increased nicotine self-administration and induced dose-dependent reinstatement, mediated by CB_1_r but not CB_2_r [95]. Likewise, Gamaleddin et al. concluded that CB_2_r were not involved in mediating nicotine self-administration or nicotine-induced seeking behavior [96].

However, a subsequent study showed that genetic deletion (CB_2_KO mice) and pharmacological blockade of CB_2_r (CB_2_r antagonism) abolished nicotine self-administration, CPP, and withdrawal. Interestingly, immunohistochemical analyses revealed that CB_2_r colocalized with nicotinic receptor α3 and α4 subunits in neurons of the NAcc and VTA. Thus, it is tempting to hypothesize that the presence of CB_2_r is necessary for the mediation of nicotine effects [81]. Accordingly, another study also found that CB_2_r is essential for nicotine-induced CPP in mice, but not for nicotine withdrawal syndrome [97]. Finally, a recent study suggested that CB_2_r located in the CA3 region of the HIP interacts with muscarinic acetylcholine receptors, modulating memory consolidation processes, a relevant aspect of drug addiction [98].

Although very little information is available, an interaction between the cholinergic system and CB_2_r may be occurring in the mesolimbic system, responsible for mediating the rewarding effects of nicotine. Considering that tobacco/cannabis smoking plus alcohol consumption tends to be very common, further studies elucidating the precise role of CB_2_r in this co-use exposure are necessary.

#### 2.3.5. CB_2_r and Neuroinflammatory Mediators Activated by Alcohol Consumption

The innate immune system has been associated with areas of the mesolimbic system and different processes involved in addictive behaviors [99,100,101]. During the last years, a growing number of studies have revealed a significant association between systemic and brain inflammation with substance use disorders [102,103], suggesting a critical role in regulating such behaviors.

Although, the functions of the immune system in defensive and repair of damage and exposure to pathogens are well established, its involvement in the regulation of brain functions, such as long-term potentiation (LTP), memory, synaptic plasticity [104,105], and emotional regulation [106], has been elucidated only recently. These physiological functions are closely related to pathways associated with the reward system and alcohol disorders.

One of the leading and most studied receptors involved in the synthesis and release of pro-inflammatory cytokines through the nuclear factor kappa B (NF-κB) pathway is toll-like receptor 4 (TLR4). Lippai et al. showed how mice deficient in this receptor (TLR4-KO) were protected from NF-κB activation induced by chronic alcohol administration [107]. In addition, the protective effect of the opioid antagonist nalmefene on the acquisition of alcohol preference in a stepwise drinking paradigm is not observed in TLR4-KO mice [108]. Interestingly, opioid agonists are postulated to non-stereoselectively activate TLR4 signaling pathways in the absence of LPS-mediated stimulation [109]. These findings highlight the involvement of the innate immune system in the mechanisms underlying the process of the development of alcohol dependence.

Concerning CB_2_r, Zoppi et al. proved the involvement of the CB_2_r in a model of stress-induced neuroinflammation. Specifically, an increase in tumor necrosis factor alpha (TNFα) mRNA expression was found in the prefrontal cortex, which decreased in mice treated with JWH-133. Interestingly, the use of genetically modified mice revealed that mice overexpressing CB_2_r in the central nervous system, CB2xP, presented reduced levels of TNFα gene expression in the PFC but increased levels in CB_2_KO mice. Similar results were shown regarding the expression of the p65 subunit of the transcription factor NF-κB. The fact that the basal level of this pathway was seen in CB_2_KO mice suggested that these mice exhibited a basal state of neuroinflammation [110].

On the other hand, Pan et al. observed a shift of microglia towards an anti-inflammatory phenotype in the hippocampus and cortex by administering the agonist JWH-133 in a pneumococcal meningitis paradigm [111]. In animal models of alcoholism, the administration of the CB_2_r agonist AM1241 promoted liver regeneration in a thioacetamide (TAA)-induced liver injury model by suppressing TLR4/miR-155/NF-κB p65 pathway [112]. Jing et al. showed similar results in a spinal cord ischemia model, where pretreatment with the CB_2_r agonist JWH-133 blocked the TLR4 MyD88/NF-κB signaling pathway [113]. Moreover, the activation of CB_2_r located on microglial cells or astrocytes in either the VTA, NAcc, or both, has been proposed to inhibit NAcc DA release indirectly by releasing cytokines and inflammatory factors, thereby inhibiting cocaine self-administration and cocaine-enhanced locomotion (Figure 1).

These studies suggest an important relationship between the CB_2_r and the processes associated with neuroinflammation in alcohol-related neurobiological and behavioral alterations. Thus, pharmacological modulation of this receptor and its potential therapeutic effects in AUD could be mediated, in part, by the improvement of inflammatory parameters.

#### 2.3.6. CB_2_r and Excitatory-Inhibitory Balance

Recent studies point out that CB_2_r are expressed in GABAergic [50,114] and glutamatergic [40,49,115,116,117] neurons. This fact suggests the relevant role that CB_2_r could play in regulating the balance of excitatory and inhibitory synaptic inputs in distinct brain regions, including those involved in drug reward. Emerging evidence demonstrates that CB_2_r activation modulates synaptic transmission in the rat medial entorhinal cortex [50], excitatory hippocampal autaptic neurons [118], and hippocampal pyramidal neurons [41]. Indeed, suppression of GABAergic inhibition could be induced by the CB_2_r agonist JWH133 and reversed by the CB_2_r antagonist AM630 [50]. Likewise, activation of CB_2_r inhibited GABA-A receptor-mediated currents [51]. Moreover, chronic treatment with a CB_2_r agonist increased excitatory transmission in glutamatergic as opposed to GABAergic synapses [41].

The firing of VTA dopaminergic neurons is tightly controlled by excitatory and inhibitory synaptic innervations. Thus, it is tempting to hypothesize that a possible modulation by CB_2_r could modify the functionality of these inputs, and therefore the reinforcing and motivational effects of alcohol. Interestingly, acute activation of CB_2_r induced a reduction in glutamatergic synaptic transmission in VTA dopaminergic neurons, including reduced presynaptic glutamate release probability. However, activation of CB_2_r induced minimal effects on either presynaptic GABA release or postsynaptic GABA-A receptor function in VTA dopaminergic neurons [45]. Similarly, another study demonstrated that VTA GABAergic neurons were insensitive to JWH133 administration [46].

In conclusion, there is limited information on how CB_2_r-mediated regulation of the excitatory/inhibitory balance may, in turn, modulate the rewarding and non-rewarding effects of alcohol in the brain. Future studies will be necessary to elucidate the underlying mechanisms.

## 3. Discussion

Despite several reviews exploring the potential role of the endocannabinoid system in AUD, mainly through the pharmacological modulation of the CB_1_r, few studies have focused on the CB_2_r. This cannabinoid receptor has emerged as a crucial target in several diseases affecting the central nervous system, including psychiatric and drug-use disorders.

Although CB_2_r occur at lower levels than CB_1_r, they can be found in neurons, microglia and astroglia in different brain regions. Interestingly, recent studies identified CB_2_r in dopaminergic, GABAergic, and glutamatergic neurons. The studies carried out to date demonstrated alterations in CB_2_r, at gene and protein level, in different brain regions of the reward system and related areas, including the AMY and HIP, by acute and chronic ethanol consumption. Moreover, changes in CB_2_r have also been found in different animal models of withdrawal and, more importantly, in mouse models evaluating the effects of stress on alcohol consumption. Later on, both pharmacological and genetic studies in rodents further suggested the involvement of CB_2_r in ethanol consumption and vulnerability since mice lacking the CB_2_r displayed increased vulnerability for alcohol consumption and motivation to drink, and treatment with CB_2_r agonists reduced alcohol consumption. Mechanistic studies pointed out the potential interactions between CB_2_r and dopaminergic, opioidergic, and cholinergic systems. Similarly, further studies are needed to elucidate the crosstalk between CB_2_r and other brain networks, including GABA and glutamatergic systems, and the involvement of microglia and astroglia in the central effects of alcohol.

## 4. Material and Methods

A literature search was conducted using the electronic databases Medline and Scopus for articles in English from the database inception to 15th November 2021. The search terms included both keywords and subject headings. The search strategy was as follows: “Receptor, Cannabinoid, CB2” AND “Alcohol-Related Disorders” AND “human/or patients”; “Receptor, Cannabinoid, CB2” AND “Alcohol” OR “ethanol” AND “rodents/or mice/or rats”.

## 5. Conclusions and Future Perspectives

Altogether, this data shows that CB_2_r functional manipulation may regulate ethanol motivation and consumption vulnerability. Interestingly, genetic and pharmacological studies revealed interactions between CB_2_r and the key targets involved in the alcohol reward system. Indeed, CB_2_r-agonism might become an approach for treating AUD that deserves further exploration. Future studies (e.g., with selective CB_2_r ligands or CB_2_r knockout and knockin mice) are necessary to evaluate ethanol rewarding aspects (e.g., conditioned place preference, voluntary consumption, or relapse). Moreover, considering the broad expression of CB_2_r in neurons, glia, and astroglia, further neurobiological studies, for instance, single-cell and RNAseq, would be interesting to characterize the mechanism of action CB_2_r-compounds at the single-cell level.

## Figures and Tables

**Figure 1 ijms-23-05908-f001:**
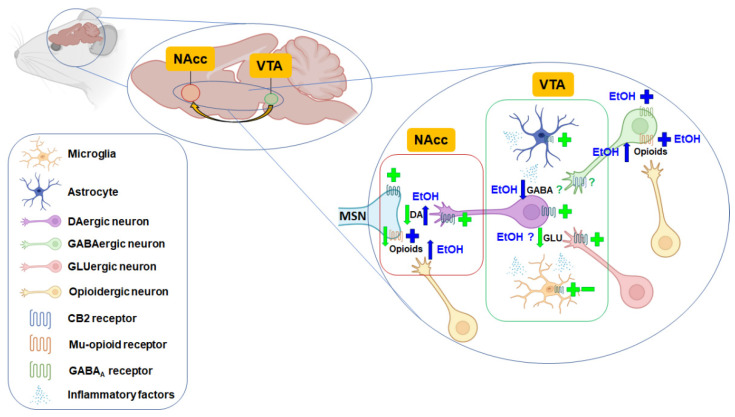
Graphic diagram illustrating the proposed mechanisms involved in ethanol actions (blue) and CB_2_r-mediated regulation of the mesolimbic DA system (green) according to studies performed in rodents. Ethanol increases the release of dopamine in the nucleus accumbens (NAcc) mainly by increasing the release of endogenous opioids acting on mu-opioid receptors and inhibiting GABAergic neurotransmission. The activation of CB_2_r induces different actions depending on its location: (1) reduction of dopamine (DA) release from ventral tegmental area (VTA) DAergic neurons projecting to the NAcc; (2) reduction of glutamate (GLU) release from GLUergic neurons projecting to VTA DAergic neurons; (3) modulation of inflammatory factors’ release from microglia or astrocytes, and (4) reduction of mu-opioid receptors density in the NAcc. According to available evidence, the role of CB_2_r in regulating GABA release from GABAergic terminals projecting to VTA DAergic neurons is not entirely clear. MSN: GABAergic inhibitory medium spiny neuron. Created with Biorender.com. +: activation/agonism; -: inhibition/antagonism; ↓: reduction; ↑: increase.

**Table 1 ijms-23-05908-t001:** CB_2_r alterations in brain regions of rodent models exposed to different regimens of alcohol addiction.

Species	Experimental Design	Results	References
Male C57/Bj6 mice	Free access to EtOH for 15 days (2–32% alcohol concentration, p.o.)	↓ *CNR2* in the midbrain of mice with higher preference for alcohol solution at a concentration of 16–32%	[60]
Male and female APPswe/PS1dE9 (AZ) and WT C57BL/6J mice	Intermittent alcohol exposure during adolescence: 2.5 g/kg, i.p., for 4 days per week, 4 weeks	↑ *CNR2* and CB_2_r protein expression in the HIP of WT mice at 6 and 12 months of age ↓ *CNR2* in the HIP of AZ mice at 12 months of age, without changes in the protein expression	[61]
Male Swiss mice	Treatment with EtOH: 2 g/kg/d, i.p., for 21 days, and mice classification as EtOH-High or EtOH-Low according to their locomotor activity after the last administration	At the end of the treatment: ↓ CB_2_r protein expression in the IL and HIP of EtOH-Low mice On the 5th day of withdrawal: ↑ CB_2_r protein expression in the Acbsh and BLA in both groups. ↑ CB_2_r protein expression in the CG1 and Acbco in EtOH-High group ↑ CB_2_r protein expression in the CeA and Dis in the EtOH-High group EtOH challenge (1.4 g/kg) after withdrawal: ↓ CB_2_r protein expression in the AMY and striatum in both groups	[62]
Male Wistar rats	Intermittent alcohol exposure during adolescence: 3 g/kg, i.p. for 4 days per week, 4 weeks	↓ *CNR2* in the striatum and HIP in adulthood	[63]
Male Wistar rats	Exposure to a two-bottle choice paradigm during adolescence: EtOH 10% (*v/v*) 24 h/4 days for 4 weeks.	↑ *CNR2* in the HIP in adulthood ↓ *CNR2* in the AMY in adulthood	[64]
Male Wistar rats	Chronic continuous ethanol diet at 10% (*w/v*) for 15 days or intermittent ethanol consumption at 10% (*w/v*), 5 days per week, 3 weeks	↓ *CNR2* in the AMY at 24 h in both groups	[65]
Male and female Wistar rats	Exposure to a chronic mild stress animal model, followed by a two-bottle choice paradigm with 20% of ethanol solution	↓ CB_2_r protein levels in the hippocampal formation of females No changes in males	[66]
Male Wistar rats	Exposure to an early restraint stress followed by an intermittent alcohol administration of 3 g/kg, 4 days a week for 4 weeks during adolescence.	↑ CB_2_r protein expression in the AMY	[67]

EtOH, ethanol; p.o., oral administration; CNR2, gene encoding for cannabinoid receptor 2; APPswe/PS1dE9, transgenic Alzheimer disease mice; AZ, Alzheimer disease; i.p., intraperitoneal administration; CB_2_r, cannabinoid receptor 2; HIP, hippocampus; AMY, amygdala; IL, infralimbic cortex; Acbsh, nucleus accumbens shell; BLA, basolateral amygdala; CG1, anterior cingulate cortex; Acbco, nucleus accumbens core; CeA, central nucleus of amygdala; Dis, dorsolateral striatum. ↓: decrease; ↑: increase.

**Table 2 ijms-23-05908-t002:** Summary of the experimental studies evaluating the influence of the genetic manipulation of CB_2_r in animal models of alcohol addiction.

Genetic Manipulation	Species	Experimental Design	Results	References
CB_2_KO	Mice	Sensitivity to acute EtOH-induced hypothermia HIC CPP two-bottle choice Oral ethanol self-administration paradigms	↓ rectal temperature ↑ HIC ↑ conditioned place preference ↑ EtOH consumption and preference in the two-bottle choice paradigm ↑ EtOH consumption and motivation	[68]
CB_2_KO	Mice	CPP	↑ conditioned place preference	[69]
CB_2_KO	Mice	Continuous and intermittent forced EtOH drinking under single and group-housing conditions	Continuous forced drinking: Individually housed CB_2_KO < grouped CB_2_KO Individually housed WT < grouped WT Intermittent forced drinking: individually housed CB_2_KO = grouped CB_2_KO individually housed WT > grouped WT	[70]
DAT-CNR2-/- conditional knockout	Mice	EtOH-induced CPP evaluation after restraint stress.	↓ conditioned place preference	[71]

EtOH, ethanol; CNR2, gene encoding for cannabinoid receptor 2; CB_2_KO, mice lacking the cannabinoid receptor 2; WT: wild-type; DAT, dopaminergic neurons in the ventral tegmental area; CPP, conditioned place preference; HIC, handling-induced convulsions. ↓: decrease; ↑: increase; <: less than; >: more than; =: as well as.

**Table 3 ijms-23-05908-t003:** Summary of the pharmacological studies in rodents studying the role of CB_2_r in animal models of alcohol addiction.

Drug	Pattern of Administration	Species	Experimental Design	Results	References
JWH015 CB_2_r agonist	20 mg/kg, i.p.	C57BL/6J mice	CPP under CMS condition	↑ Voluntary ethanol consumption	[60]
β-caryophyllene CB_2_r agonist	25, 50 and 100 mg/kg	C57BL/6J mice	CPP VC	↓ CPP ↓ Voluntary ethanol consumption and preference	[72]
JWH133 CB_2_r agonist	10 and 20 mg/kg, i.p.	HS/Ibg mice	CPP VC	No differences	[69]
JWH133 CB_2_r agonist	5 mg/kg, i.p.	C57BL/6J mice	CPP	↓ CPP	[73]
JWH133 CB_2_r agonist	1 mg/kg, i.p.	C57BL/6J mice	OESA	↓ Motivation to drink ethanol	[74]
JWH133 CB_2_r agonist	1 mL per 100 g of body weight, i.p.	C57BL/6J mice	CPP	↓ Motivation to drink ethanol	[75]
JWH133 CB_2_r agonist	0.2mg/kg, i.p.	Wistar rats	Free access EtOH 10%, 2 weeks	↑ number of BrdU+ cells in the subventricular zone of the DG	[76]
AM630 CB_2_r antagonist	3 mg/kg, i.p.	C57BL/6J mice	CPP under CMS condition	↓ Motivation to drink ethanol	[60]
AM630 CB_2_r antagonist	10 and 20 mg/kg, i.p.	HS/Ibg mice C57BL/6J mice	CPP CV	No differences	[69]
AM630 CB_2_r antagonist	1 mg/kg, i.p.	C57BL/6J mice	OESA	↑ Motivation to drink ethanol	[74]

CPP: conditioned place preference, VC: ethanol voluntary consumption, OESA: oral ethanol self-administration, CMS: chronic mild stress. ↓: decrease; ↑: increase.

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
