# Peer review of "Role of Cannabinoid CB2 Receptor in Alcohol Use Disorders: From Animal to Human Studies"

_ijms, 2022, doi:10.3390/ijms23115908_

Round 1

Reviewer 1 Report

The authors systematically include in the manuscript all the information from CB2 and Alcohol studies, with the prevalence of basic science work. The topic is very important and the manuscript will help new scientist that are approaching the topic to have a clear and comprehensive knowledge of it.

Very good organization of the tabs, and figures.

Minor comments:

-Why was not EtOH included in the research strategy?

-There is a paragraph on CB2R and the opioid system, and one on the Glutamate and GABA balance, but a session on the cholinergic system is missing. Nicotine, Cannabis and Alcohol are often used together, it would be interesting to understand the role that CB2R mediate in this specific co-use exposure.

-Please insert in the introduction a specific description on the type of CB2 receptors and the cellular mechanism that their activation comport. 

Author Response

Please find enclosed a point-by-point response to your comments. 

Reviewer 2 Report

Dear Editor,

            In the present manuscript, the authors reviewed the role of CB2 receptor in alcohol-driven disorders in animals and humans. In general, the manuscript has been well written. Here are my comments:

Abstract

  1. Page 1, line no 21: In addition, mice lacking CB2r (CB2KO) display alcohol consumption, motivation, and relapse alterations. If applicable, it would be better if the authors mention it properly. I mean whether the CB2r knockout mice were more prone to the development of alcohol-based disease or vice versa. Please mention that.
  2. Page 1, line no. 27: Conclusions: This review includes a conclusion remarking on the future role………. I would suggest removing ‘This review includes a conclusion’ as already it comes under the subdivision ‘Conclusions’. Revise the remaining sentence accordingly.
  3. Page 1, line 28: ……… clinical studies to evaluate further CB2r…… Change it as …. clinical studies to further evaluate CB2r……

Results

  1. Page 2, line no. 80: I am not sure having subtitle ‘Results’ is OK in a review article. I would rather suggest omitting this and starting with 2. Evidence from rodent studies.
  2. Page 2 line no. 83: ……..several animal models modulating different stages of……… Replace ‘modulating’ with any other suitable word.
  3. I would suggest inserting the above sentence as the last sentence in that paragraph as it is not looking good to talk about the table in the very first sentence. This is applicable for all the tables.
  4. Page 2, line no 86: In contrast, no alterations were found in mice with lower preference. This sentence can be explained in detail.
  5. Page 3, line 89: Table 1. CB2r alterations in brain regions of rodents exposed to different animal models of alcohol addiction. This sentence can be modified as follows: Table 1. CB2r alterations in brain regions of rodent models exposed to different regimens of alcohol addiction.
  6. Make sure there are no typo errors in the table. What is p.o.? Kindly elaborate all the abbreviations used in the table legend.
  7. In table 1, CB2r protein levels in the hippocampal formation of females (60). What is hippocampal formation? What about males in that study?
  8. Page 4, line 105: ……neuroinflammatory mediators….Mention those mediators observed in that study.
  9. Page 4, line 107: A recent study developed by Ledesma et al. [57] aimed to analyze the impact of ethanol binge drinking during adolescence on cognitive disturbances showed that preexposure..... I would suggest revising the sentence as follows, for ex., A recent study developed by Ledesma et al. [57] aimed to analyze the impact of ethanol binge drinking during adolescence on cognitive disturbances. This study showed that preexposure….
  10. Page 4, line 111: …..mice model of Alzheimer's disease, APPswe/PS1dE9 mice, there was a tendency for CB2r gene expression to be reduced at 6-months of age, being statistically significant at 12- months age. I would suggest revising the sentence as follows, for ex., …..mice model of Alzheimer's disease, APPswe/PS1dE9 mice, CB2r gene expression was reduced at 6-months of age while statistical significance was noticed at 12-months. Please ensure such kind of modifications throughout the manuscript.
  11. Page 4, line 129: The authors conclude… Change it as The authors concluded….
  12. Page 4, line 135: Taken together, CB2r down-regulation appears to be related to resilience to develop motor sensitization [58]. This sentence can be changes as follows, for ex., Taken together, the results showed that CB2r down-regulation was appeared to be related to the development of motor sensitization [58].
  13. Page 4, line 147: ……CB2r gene expression in the AMY, pointing out its potential implication in negative motivational states and enhanced stress responsivity….. The sentence can be changes as follows, for ex., ………….CB2r gene expression in the AMY, which pointed out its potential implication in negative motivational states and enhanced the stress responsivity…..
  14. Page 4, line 164: ….suggesting that several factors as the strain of rodent,…. I believe it should be ………suggesting that several factors such as the strain of rodent,….
  15. Page 4, line 167: Moreover, advanced techniques as……. I believe it should be Moreover, advanced techniques such as……
  16. Page 6, line 224: Like last section, it would be nice if the authors would add the summary notes in the end for all sections.
  17. Page 6 line 228: First studies revealed….. Should it be studies or study?
  18. Page 7 line 247: ……pharmacological regulation of rCB2 on alcohol consumption……. rCB2 ?
  19. Page 8, line 277: …… showing a reduction in both regions. As the data has not been published yet, it would not be good to say ‘showing’. Please replace that with another word.
  20. Page 13, line 504: Moreover, and considering the broad expression…. Should it be Moreover, considering the broad expression….?
  21. Page 13, line 505: ……..for instance, single-cell and RNAseq, would… Should it be ……… for instance, single-cell RNAseq, would…..?
  22. Page 13, line 501: ……. CB2r-antagonism appears to be the most promising approach…… Should it be CB2r’agonism?

Figures

What do green plus and minus symbols denote in the figure?

Author Response

Please, find enclosed a point-by-point response to your comments. 
